# A Silicified Carboniferous Lycopsid Forest in the Colorado Rocky Mountains, USA

**Mike Viney** [1] , **Robert D. Hickey** [2] **and George E. Mustoe** [3,*]

1.  College of Natural Sciences Education and Outreach Center, Colorado State University, Fort Collins, CO 80423, USA; mike.viney@colostate.edu
2.  Arkansas Headwaters Recreation Area State Park, Salida, CO 81201, USA; r.c.hickey75@gmail.com
3.  Geology Department, Western Washington University, Bellingham, WA 98225, USA
*   Correspondence: mustoeg@wwu.edu

**Abstract:** The 1930 discovery of Carboniferous lycopsid fossils in south central Colorado resulted in the naming of a new species of scale tree, *Lepidodendron johnsonii* (=*Lepidophloios johnsonii* (Arnold) DiMichele). Cellular structures of *L. johnsonii* axes and periderm are preserved in silica—an unusual mode of fossil preservation for Pennsylvanian lycopsid plant remains. The early reports on the Trout Creek lycopsid fossils focused on taxonomic and paleobotanical aspects. Our 2019 reinvestigation of the locality produced many new specimens and a wealth of new data from a variety of analytical methods. Optical microscopy, X-ray diffraction, scanning electron microscopy, energy dispersive electron spectroscopy, determination of specific gravity, and Loss on Ignition provide details of mineralization. Cell walls are preserved with very small fine quartz particles, and cell lumina are filled with microcrystalline quartz. Some cell exteriors are encrusted with euhedral quartz crystals. These multiple forms of quartz are evidence that petrifaction involved several episodes of silicification. The dark color of the fossil wood and siliceous matrix appears to be caused by traces of dispersed carbon, but 500 °C Loss on Ignition reveals that the fossil wood preserves only very small amounts of the original organic matter.

**Keywords:** Carboniferous; Colorado; fossil wood; *Lepidodendron*; *Lepidophloios*; lycopsid; paleobotany; silicification

## 1. Introduction

Lycopsids evolved as large arborescent forms, attaining maximum size during the Carboniferous, when they became one of the dominant members of low elevation swamp forests (Figure 1). Mature trees exceeded 30 m, with trunk diameters of 1 meter or more. Arborescent lycopsids disappeared by the close of the Paleozoic, but herbaceous forms persist to the modern era. Extant genera *Lycopodium* and *Isoetes* rarely exceed 0.75 m in height. The taxonomy of arborescent lycophytes merits explanation. Trunk tissues bear distinctive leaf scars that allow various genera to be identified (e.g., *Lepidodendron*, *Pleuromeia, Lepidophloios*). Underground rooting structures are known by the genus name *Stigmaria*. Dispersed leaves are identified as *Lepidophylloides*. Intact reproductive organs are assigned to the form genus *Lepidostrobus or Flemingites*; fertile leaves that make up these cone-like structures are named *Lepidostrobophyllum*.

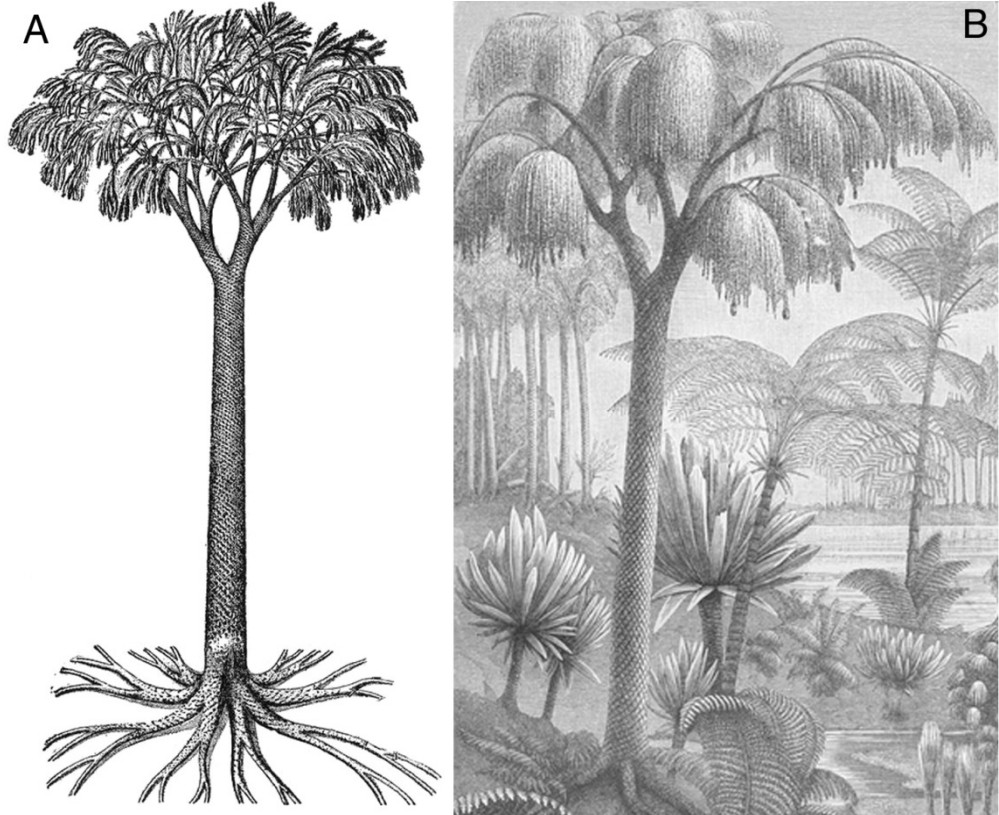

**Figure 1.** Carboniferous lycopsids. (**A**) *Lepidodendron obovatum*. Drawing adapted from [1]. (**B**) Coal swamp reconstruction showing arborescent lycopsids as the largest trees, circa 1890 etching, public domain.

During the summer of 1930, geologist J. Harlan Johnson discovered remnants of an early Pennsylvanian fossil forest near Trout Creek Pass, in Chafee County Colorado, USA (Figure 2). Botanical remains consist of numerous fragments of fossilized *Lepidodendron johnsonii (= Lepidophloios johnsonii* (Arnold) DiMichele) lycopsid trunks. No other genera have been found at this locality. These fossils are evidence of ancient trees that had diameters as large as 0.6 m. The lycopsid remains were studied by Arnold [2], who provided a detailed description of the site and a careful analysis of the anatomy of the silicified tissue. Arnold named the material as a new species, *Lepidodendron johnsonnii*. Subsequently, the Trout Creek Pass fossils were included in various paleobotany textbooks (e.g., [3,4]). The taxon has since been renamed as *Lepidophloios johnsonii* (Arnold) DiMichele [5].

Our decision to reevaluate the Trout Creek Pass fossil locality was based on several factors. Arnold's original 1940 report [2] is rich in detail, but the investigations focused on specimens that showed exceptionally good cellular preservation, which provided a basis for taxonomic studies. Arnold believed that these trunks had been preserved in an upright position, an interpretation that is reflected in his reconstruction of the landscape, showing numerous standing trees. Our field studies revealed that the well-preserved stems illustrated by Arnold are relatively rare, and that the majority of specimens represent trunk fragments that were preserved in a horizontal position. Rather than originating as a standing forest, the paleoenvironment may instead have been a floodplain that preserved an abundance of fluvially transported wood fragments.

A second reason for our study is that knowledge of the geologic origin of the Colorado Rocky Mountains has improved considerably since 1940, the later awareness of the importance of plate tectonics being a particularly relevant example.

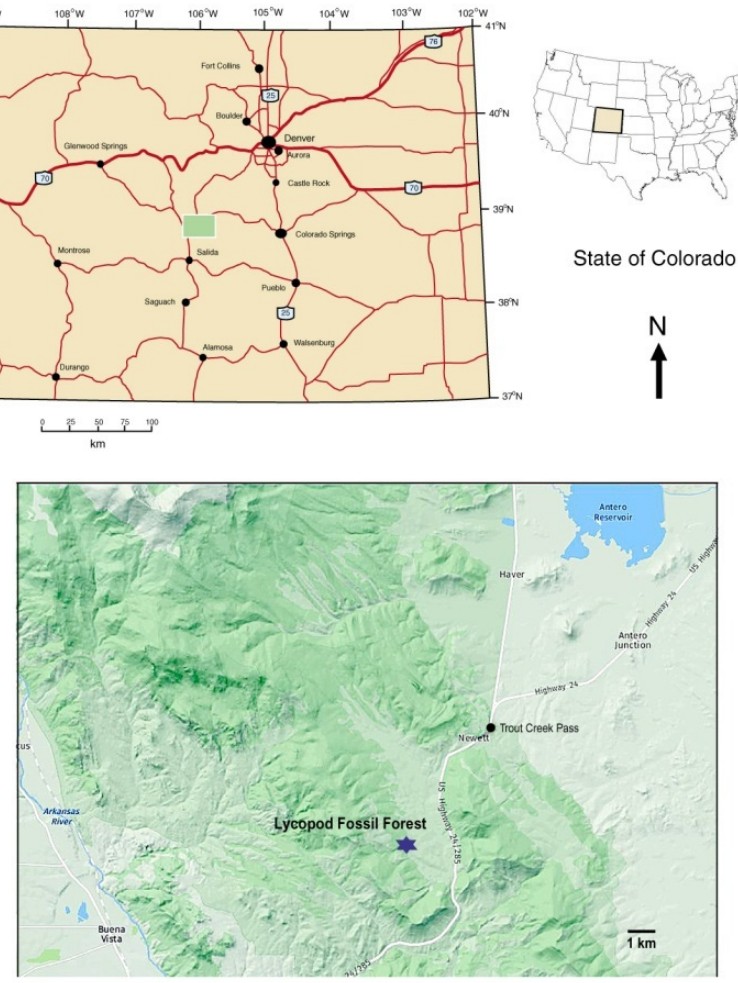

**Figure 2.** Location Map.

Finally, although the quality of the original taxonomic studies is excellent, the work was based on transmitted light microscopy. This method provides detailed anatomical information, but very little evidence to understand the mineral processes that caused the wood to become fossilized. This issue is important, because the Trout Creek fossil lycopsid wood is mineralized with microcrystalline quartz. Lycopsid stem and root structures are abundant in other locations, principally preserved in three ways: as silt or sand casts (internal and external), permineralized with calcium carbonate in coal balls, and as compressions/impressions in shale and concretions. The Trout Creek fossil locality therefore appears to represent unusual conditions of fossilization. We have used polarized light optical microscopy, scanning electron microscopy, X-ray diffraction, density determination, and 500 °C loss on ignition to investigate the mineralogy in detail.

## 2. Site Description

The lycopsid fossil study site is located in the east central part of Chaffee County, CO, in the central portion of the Mosquito Range, just west of U.S. Highway 24/285. The site is on Colorado State Trust Land, and a permit is required for specimen collecting. Weathered specimens ranging in size from small hand samples to small boulder size can be found lying on the surface throughout a large meadow area. Dry creek beds to the north of the meadow contain eroded silicified specimens.

## 3. Geologic Setting

The Trout Creek fossil locality provides a glimpse into the complex geologic history of the Rocky Mountain region. We are discussing the tectonic history of the site in detail to explain how the remains of a lowland swamp came to be preserved in a high mountain meadow (Figure 3).

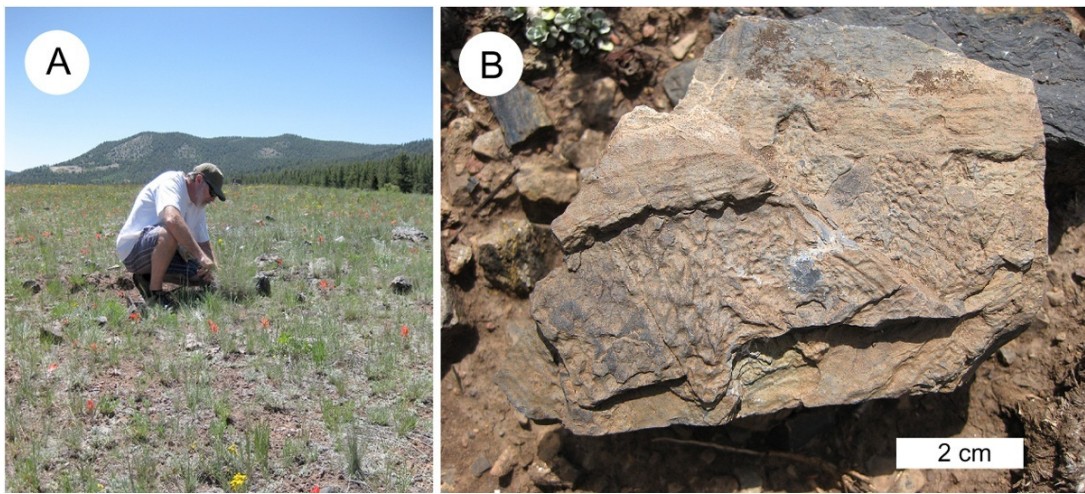

**Figure 3.** Trout Creek lycopsid fossil site. (**A**). Bill Evans examines the fossil specimens that occur as surface material. (**B**) Irregular shaped specimen showing layers of compressions/impressions exhibiting leaf scars oriented in different directions.

The Trout Creek fossil locality occurs in an expansive wildflower meadow that lies at an elevation of 2800 m (9200 ft) in the southern Rocky Mountains. This setting is markedly different from the paleogeographic conditions that existed when the lycopsid forest was alive. One important factor involves global tectonics [6]. During the early Paleozoic, breakup of the supercontinent Pannotia resulted in smaller continents that included Laurentia (composed of North America and Greenland), Gondwana (South America, Africa, Antarctica, India, and Australia), Baltica (Europe) and Siberia. The edges of these new continents were typically passive marginal basins, which, accompanied by sea level rise, caused continental interiors to be flooded by shallow seas.

This global geography set the stage for the development of the Trout Creek fossil forest. An important step was the collision between Gondwana and Laurentia during the middle Carboniferous to early Permian, causing extensive orogeny in North America (Figure 4). In the early Pennsylvanian, the onset of uplift caused marine regression, leaving behind the shale, sandstone, and limestone of the Belden and Glen Eyrie Formations of Central and Eastern Colorado, respectively. These sediments are exposed in the southern Mosquito Range at Trout Creek Pass. This sea level retreat marked the end of more than 200 million years of marine sedimentation in Colorado. The newly emerged lowlands provided a favorable habitat for coastal swamps, including the Trout Creek site. A few other Pennsylvanian plant sites have been reported from central Colorado, but these occurrences primarily consist of foliage impressions [7,8].

Early Pennsylvanian regional uplift led to the rise of the Ancestral Rocky Mountains (ARM), which comprised two island massifs: Frontrangia, in what is now central Colorado, and Uncompaghria to the southwest. The bedrock source for these ranges included Precambrian rocks, and Paleozoic limestone and dolomites. Although the ARM rose to elevations of approximately 3000 m (10,000 ft) by 300 Ma, within 50 Ma the mountain range had been largely obliterated by erosion, producing extensive alluvial fans that extended into adjacent floodplains. By the close of the Permian, the ARM had been almost completely eroded, their bedrock foundation now buried by their own debris. In the late Paleozoic, the Laurentia/Gondwana contact zone lay close to the equator, with tropical and semitropical climate-favored development of lush plant growth in the coastal lowlands, producing extensive coal

swamps. In the late Paleozoic Era, these inland seas retreated, and regions that had once received thick limestone reef deposits were now sites of deposition for fluvial sand and shale, with organic debris locally abundant.

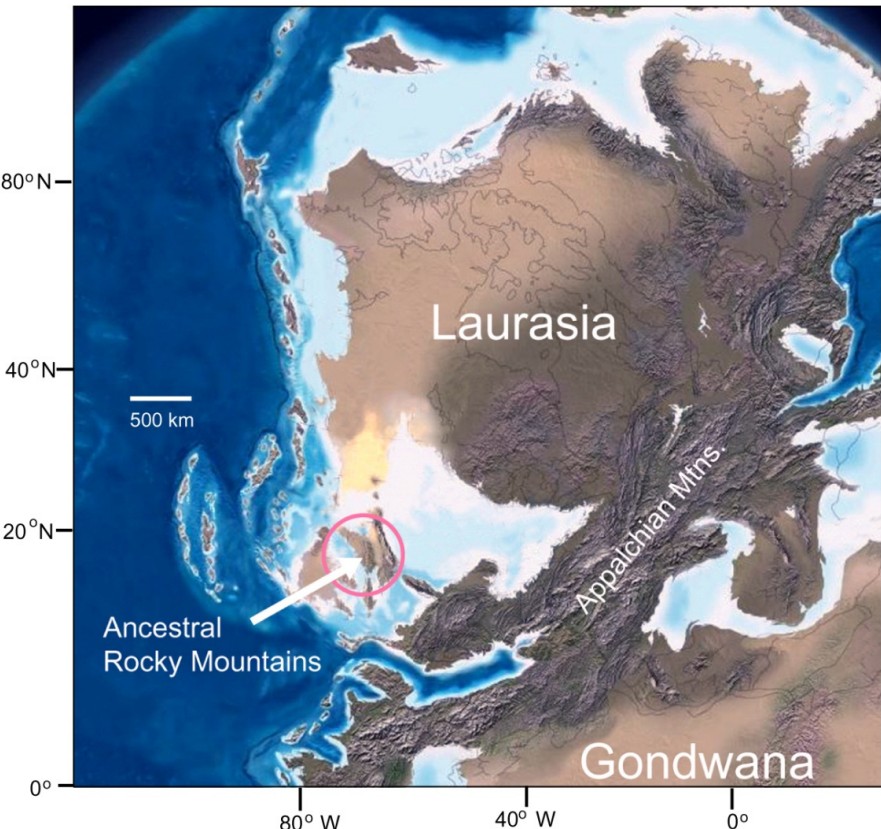

**Figure 4.** The collision of the supercontinents Laurasia and Gondwana resulted in the rise of the Appalachian Mountains, and the appearance of the Ancestral Rocky Mountains as offshore islands. This illustration represents Pennsylvanian tectonic positions at approximately 300 Ma.

The present Rocky Mountains are a result of the Laramide Orogeny, when late Cretaceous and Paleocene faulting brought crystalline rocks to the surface to form tilt-blocks and anticlinal ridges [9]. Erosion of these highlands led to the accumulation of Eocene sediments in adjacent basins. Lesser uplift events continued during the Miocene and Oligocene, accompanied by pulses of sedimentation. During this period, North America became recognizable as a discrete continent, with rotation and northward transport (Figure 5). A detailed description of the regional geology is provided in the supplementary material.

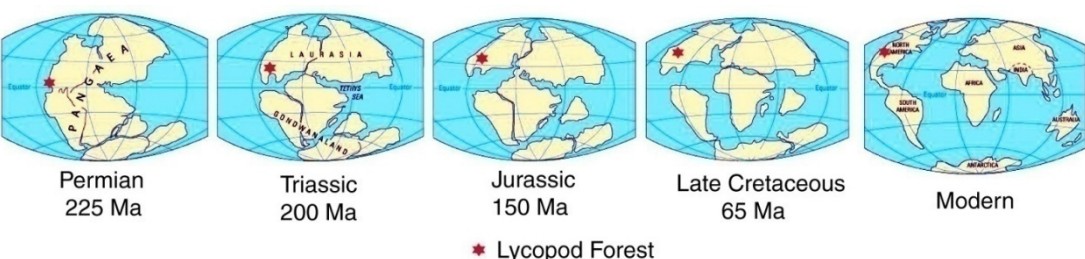

**Figure 5.** Map series showing relative latitude of lycopsid fossil forest over time. Figure adapted from [10].

This tectonic history explains how the Trout Creek fossil forest originated as a lowland swamp at low latitude, and why the plant remains are now found in a high alpine meadow far to the north of the original location. This complex geologic history can be understood in more detail from the strata exposed in the present vicinity of the Trout Creek Pass.

The Trout Creek fossil site lies within a thick sequence of Paleozoic sedimentary rocks that overlay Proterozoic granitic basement rocks. The generalized stratigraphy is shown in Figure 6. The regional structure was deformed, primarily as a result of the late Cretaceous to early Tertiary Laramide Orogeny. Individual Paleozoic Formations are typically separated by unconformities, evidence that they were produced by discrete episodes of sedimentation that were separated by prolonged intervals of non-deposition. The Belden Shale Formation, which contains the lycopsid fossils, is poorly exposed in the study area because of the susceptibility of the sediment to weathering. The rock consists of dark gray, brownish gray, and black shale. The general thickness of the strata is estimated to be 260 m (850′), but the Belden Shale reaches a maximum thickness of 520 m (1700′) to the north of the fossil forest site. The formation has a transitional contact with the overlying Minturn Formation, which consists of sandstone and shale, with lesser amounts of siltstone, limestone, and dolomite. The combination of clastic and carbonate beds in the Belden Shale and Minturn Formations are evidence of Pennsylvanian transgression and regression associated with the tectonic unification of Laurentia and Gondwana. The Belden Shale Formation contains marine invertebrate fossils and terrestrial plants, occurrences that suggest that the lycopsid forest occupied a coastal lowland.

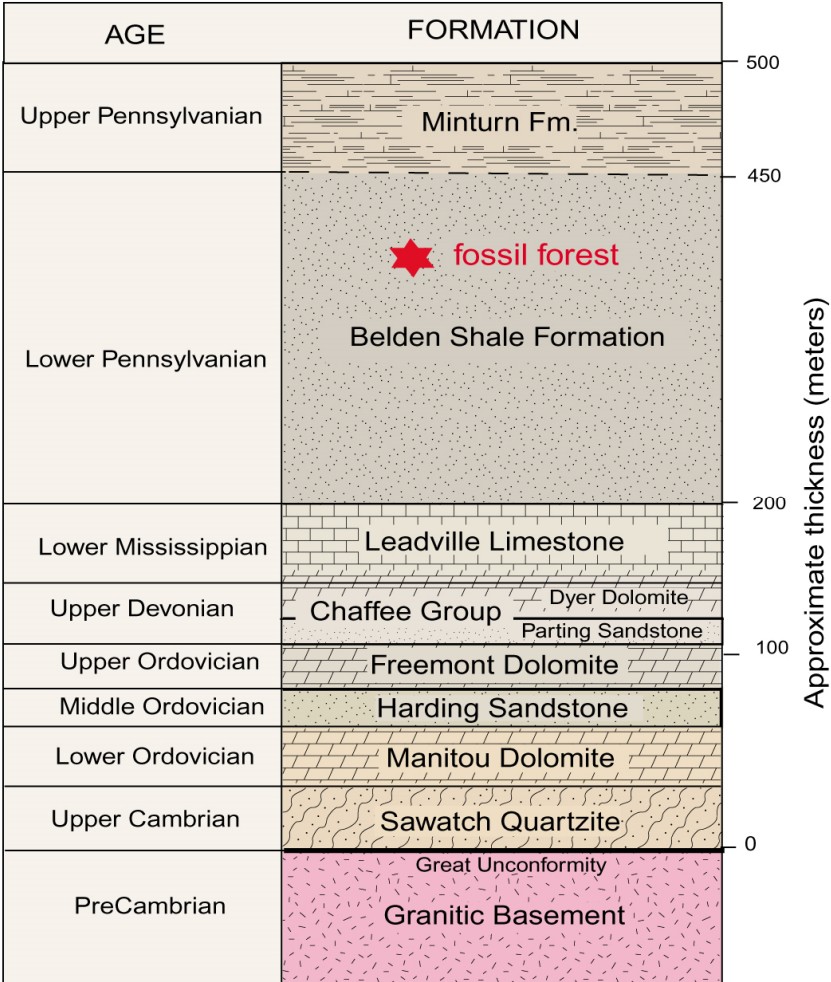

**Figure 6.** Generalized stratigraphy. A detailed description of the regional geology is provided in a separate data file.

## 4. Previous Research

Lycopsids were among the first large trees in the Earth's history, preceded only by the Late Devonian *Archaeopteris*. They were the tallest members of Carboniferous coal swamps. Lycopsid remains are abundantly preserved as casts and in in calcareous coal balls, where fossilized organs include wood, rootlets, and foliage. Because of their evolutionary importance, lycopsid fossil have been the subject of considerable research. These studies have covered a vast scale, a span that ranges across entire fossil forests [11,12], individual trunks [13], root networks [14,15], and spores [16]. Arborescent lycopsids were well known to paleontologists by the late 19th century (e.g., [17–19]), but the 1930 discovery of the Trout Creek fossil locality captured attention because of the silica mineralization, a contrast to the calcareous preservations that had previously been studied. In addition, the excellent anatomical preservation provided an opportunity to better understand the cellular architecture. The site was first mentioned in print by Johnson in 1932 [20]; in 1940, Arnold [2] produced a monograph that is notable for its detailed description of geological as well as paleobotanical characteristics of the Trout Creek fossil locality. For our report, we are not providing a detailed summary of the anatomical characteristics of the silicified tissues. Instead, readers are referred to the original description [2] and two later summaries [3,4]. General characteristics are reviewed in this paper as a foundation for understanding the microscopic images and discussions of fossilization processes.

In 1938, Gould and Johnson visited the site to examine and collect fossil material [3]. Fossils were found to consist of portions of silicified trunks and numerous shale fragments that preserved imprints of *Lepidodendron* and *Lepidophloios*. Johnson reported that most of the cell structure breaks down when treated with hydrofluoric acid, evidence that these silicified tissues preserved very little relict organic matter. As a result, cellulose acetate peels, such as are used for studying coal balls, were found to have little value. Arnold used detailed observations of thin sections representing axes to describe a new species of lycopsid, *Lepidodendron johnsonii*, named in honor of J. Harlan Johnson, who announced the discovery of the Trout Creek fossil deposit in 1932. The mineralized stem section collected by Arnold consists of a pith area, primary xylem, secondary xylem, and some cortex tissue. The phloem, outer cortex tissue and periderm are not present in this specimen (Figure 7). Like most axes found by Arnold at the site, the pith area is filled with debris and structures he identified as *Stigmaria* roots. Arnold interpreted the presence of *Stigmaria* roots in the pith as evidence that the roots of lycopsid trees grew around and through dead trunks that were buried within the soil. We did not observe roots in specimens that we collected. The poor preservation of the pith and cortex are evidence of rapid decay of these poorly lignified tissues following the death of the lycopsid. Even the well-lignified periderm often exhibits some decay.

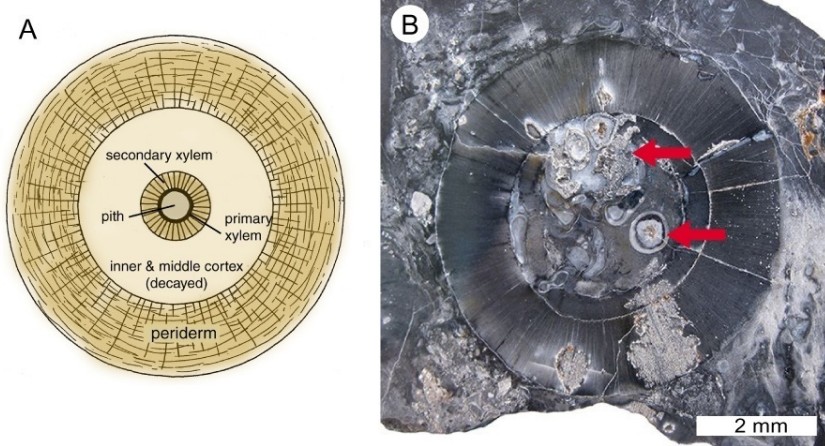

**Figure 7.** Lycopsid stem anatomy. (**A**) Transverse view of lycopsid stem, showing a small central axis composed of pith, primary xylem, and secondary xylem (wood). The small central axis is surrounded

by a thick outer bark composed of inner and outer cortex enveloped by fibrous periderm. The periderm provided the structural stability for the lycopsid tree. (**B**) Central axis of *L. johnsonii* specimen collected by Arnold, now at Colorado School of Mines Museum of Geology, Golden, CO USA. The cortex, which originally comprised up to 80–90% of the trunk volume, is not preserved. The primary xylem ring is partially preserved, whereas the secondary xylem ring is intact. Arrows denote circular structures that Arnold interpreted as lycophyte rootlets (*Stigmaria*) that invaded the decayed pith zone prior to fossilization.

## 5. Materials and Methods

Scanning electron photomicrographs were obtained using a Vega3 SEM equipped with an Oxford X-ray detector running AzTek software. Optical photomicrographs were taken using a five megapixel digital microscope camera mounted on a Zeiss petrographic microscope. X-ray diffraction patterns are from a Rigaxu Geigerflex diffractometer, using nickel-filtered Cu K$\alpha$ radiation. Loss on ignition measurements were made by measuring the weight loss of approximately 5 g of powdered fossil wood after heating for eight hours at 500 °C in an electric laboratory furnace. Samples were heated overnight at 100 °C to remove adsorbed moisture prior to LOI determination.

## 6. Results

### 6.1. Mineral Features

Specimens at the Trout Creek fossil locality typically consist of lycopsid trunk fragments whose exterior surfaces preserve the texture of the stem's leaf-scarred outer layer (Figure 3B). The interiors of these specimens are variable, but the fossils typically represent a sedimentary matrix that preserves a wood-shaped space, enclosing an internal filling that consists of three basic types:

1. Anatomically well-preserved lycopsid stems and stem fragments (Figure 8A). These are the materials that have been described by previous investigators [2–4,20]. The cortex, which originally comprised up to 80–90% of the trunk volume, is not preserved. Only the lignified cells, making up the small xylem ring of the central axes and portions of the outer periderm, are well-preserved in silica;
2. Wood fragments that show evidence of gravitational settling. This material consists of silicified wood that owes its dark color to relict organic matter, and specimens that have been bleached to a light color (Figure 8C,D). These specimens contain mineralized tissue fragments that occupy only a portion of the original volume of the wood, as indicated by the outer cast surface. Thin sections show that the remaining space is typically filled with microcrystalline quartz that precipitated from Si-bearing groundwater, as evidenced by the absence of cellular remains in these regions;
3. Mixtures of microcrystalline quartz and coalified organic matter, commonly showing intense deformation (Figure 8D,E).

The casts do not contain visible clastic sediment, suggesting that, during diagenesis, the outer cast surface remained as an intact barrier that prevented the influx of solid material from the surrounding environment.

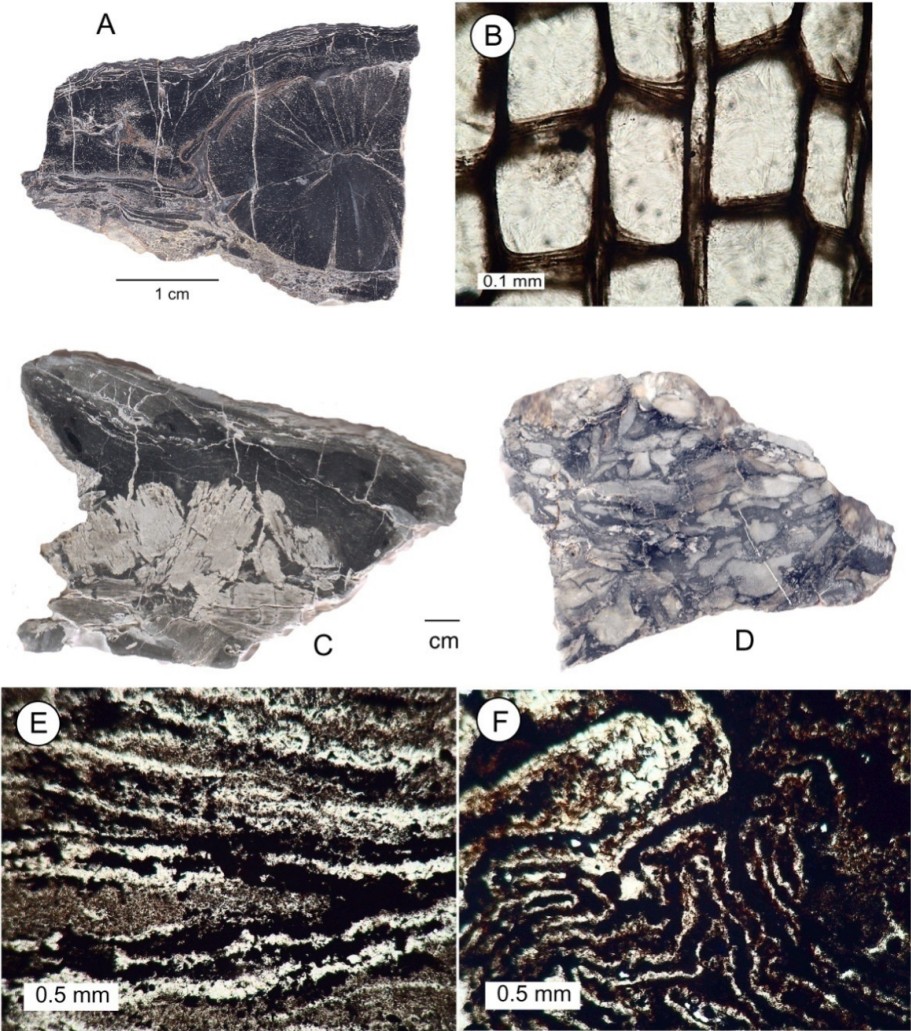

**Figure 8.** Examples of various types of preservation of fossil lycopsid wood. (**A**) Small central lycopsid axis with secondary xylem. (**B**) Transverse view of tracheids in secondary xylem. (**C**) Layers of bark compression/impressions associated with fragmented pieces of silicified wood or fiber oriented in different directions, naturally bleached, and showing signs of gravitational settling. (**D**) Fragmented pieces of lycopsid tissue that have settled to the bottom of a space created when the original wood mass deteriorated. (**E**,**F**) Photomicrographs of thin sections of carbonaceous sediment created by intense deformation of coalified lycopsid wood. Dark, carbon-rich layers are interspersed with thin bands of crystalline quartz.

*6.2. Optical Microscopy*

Specimens show abundant evidence of quartz mineralization. Specimens that show a well-preserved cell structure contain a microcrystalline quartz filling cell lumina (Figure 9). Cell walls are typically opaque in polarized light views. There are three possible explanations for this. One is that the cell walls are composed of relict organic matter. This interpretation conflicts with loss on ignition data that reveal that relict carbon has low abundance in the fossil wood (Table 1). A second possibility is that the cell walls are mineralized with amorphous opal (opal-A). The Lower Pennsylvanian age of the deposit makes an opaline composition unlikely, because opal-A transforms relatively rapidly to opal-CT, and from there to quartz; opalized wood is common in Cenozoic formations, but virtually unknown in older deposits [21]. We prefer the third interpretation, which is that the cell walls appear to be isotropic because they consist of aggregates of extremely small quartz particles. Because birefringence is related to specimen thickness, in 30 micron thick petrographic slides, the cell walls appear dark under

polarized light because the individual quartz particles are so small that they do not show interference colors. This interpretation is supported by SEM images that show the microstructure of cell walls (Figure 10).

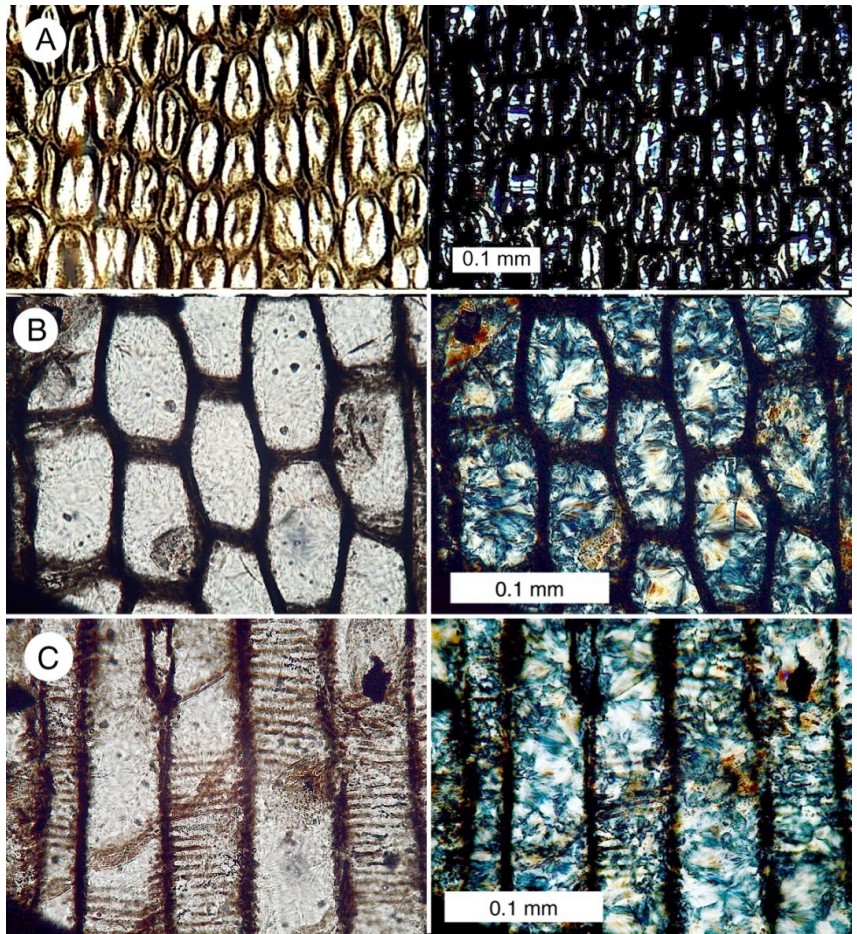

**Figure 9.** Optical photomicrographs of lycopsid wood. In each pair, the photo on the left is made with ordinary transmitted light; the right photo shows the polarized light view. Birefringence colors in polarized images are indicative of quartz/chalcedony. (**A**) Transverse view of the fiber-like cells that form periderm. Dark matter at the center of these cells and oriented in a radial direction was interpreted by Arnold as innermost wall material that had detached during degradation. (**B**) Transverse view of tracheids in secondary xylem (**C**) Tangential view of secondary xylem, showing scalariform tracheids.

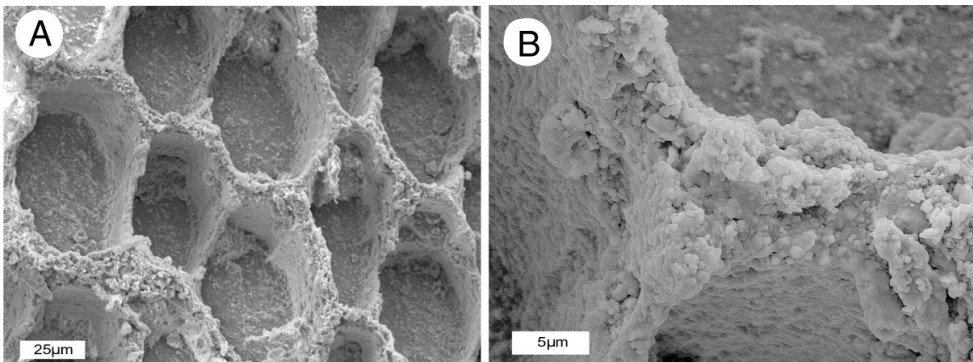

**Figure 10.** SEM images showing oblique transverse view of periderm cells. (**A**) Cell walls stand out in high topographic relief. (**B**) Close-up views show cell walls have been replaced by an aggregate of very small quartz grains.

The dark color of the cell walls is not caused by traces of amorphous compounds such as iron and manganese, based on the absence of characteristic peaks for these elements in SEM/XRF spectra.

The thin section views show that chalcedony and crystalline quartz fill larger spaces within the fossil wood, including rot pockets and open fractures (Figure 11).

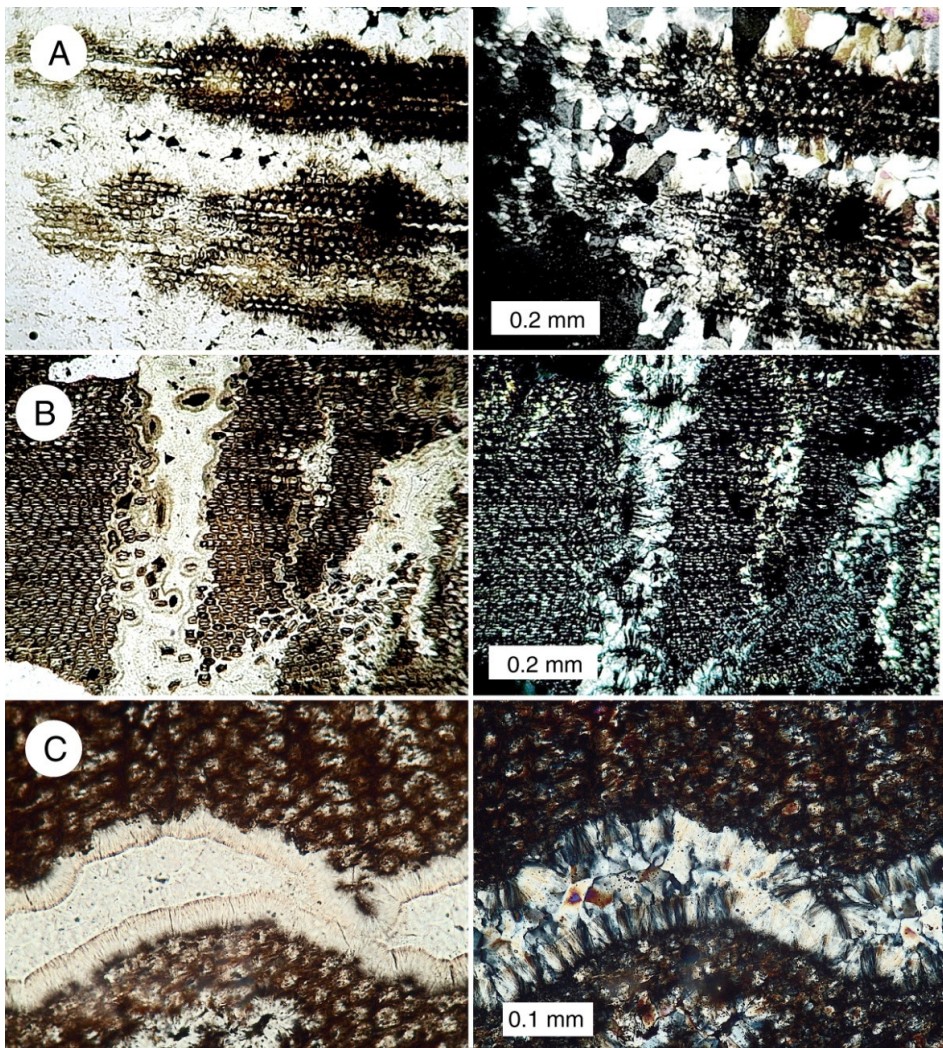

**Figure 11.** Optical photomicrographs of 30 micron thick slices of periderm tissue, transverse orientation. Images on the left are ordinary transmitted light; right images are polarized light views. (**A**) Degraded tissue where individual cells are filled with chalcedony/quartz. Voids created by decay have been filled with microcrystalline quartz. (**B**,**C**) Veinlets cross-cut cell layers. Each veinlet consists of thin margins of chalcedony with microcrystalline quartz filling the interior zone.

**Table 1.** Density and 550°C Loss on Ignition.

| Sample | Density g/cm$^3$ | Weight Loss after 550 °C |
|--------|------------------|--------------------------|
| CL-1   | 2.42 | 0.55% |
| CL-2   | 2.54 | 0.57% |
| CL-3   | 2.32 | 0.82  |
| CL-4   | 2.46 | 1.23% |
| CL-5   | 2.56 | 0.83% |
| CL-6   | 2.31 | 0.72% |
| CL-8   | 2.59 | 0.76% |
| CL-A1  | 2.54 | 1.56% |
| L-BLK  | 2.58 | 1.69% |

### 6.3. Density and 500 °C Loss on Ignition

The density of silicified wood can be used to evaluate the mineralogy. Woods mineralized with opal have densities of 1.9–2.1 g/cm$^3$, compared to 2.3–2.6 g/cm$^3$ for wood mineralized with chalcedony or quartz [22]. These measurements are only valid for samples that are solidly mineralized; open spaces reduce the measured density.

Most Colorado lycopsid specimens have densities of 2.42–2.59 g/cm$^3$, indicative of quartz/ chalcedony (Table 1). The exceptions are CL-3, and CL-6, where slight porosities explain the reduced density.

### 6.4. Scanning Electron Microscopy

SEM images provide topographic information for specimen areas that are not solidly mineralized. Figure 10 shows a specimen surface where the cell walls of periderm stand in relief compared to the quartz-filled lumina. At high magnification, the cell walls can be seen to consist of a randomly oriented aggregate of very small silica particles, the composition evidenced by EDS spectra. The particulate texture suggests the possibility that the cell walls were initially mineralized with opal that has transformed to quartz during diagenesis. The apparent isotropic character of cell walls viewed under polarized light is presumably caused by the small grain size, where individual quartz particles are so small that petrographic thin sections do not display the normal quartz birefringence colors that would be expected for a 30 micron on a thick layer.

Longitudinal SEM images clearly show the microcrystalline quartz that has replaced wood tissue (Figure 12). Cell interiors (lumina) contain solid crystalline masses, with euhedral crystals encrusting outer cell surfaces. The sizes of individual crystals are variable, ranging from less than one micron to more than five microns. The larger crystals commonly developed on a substrate of small crystals (e.g., Figure 12C), evidence that quartz precipitation occurred in several episodes.

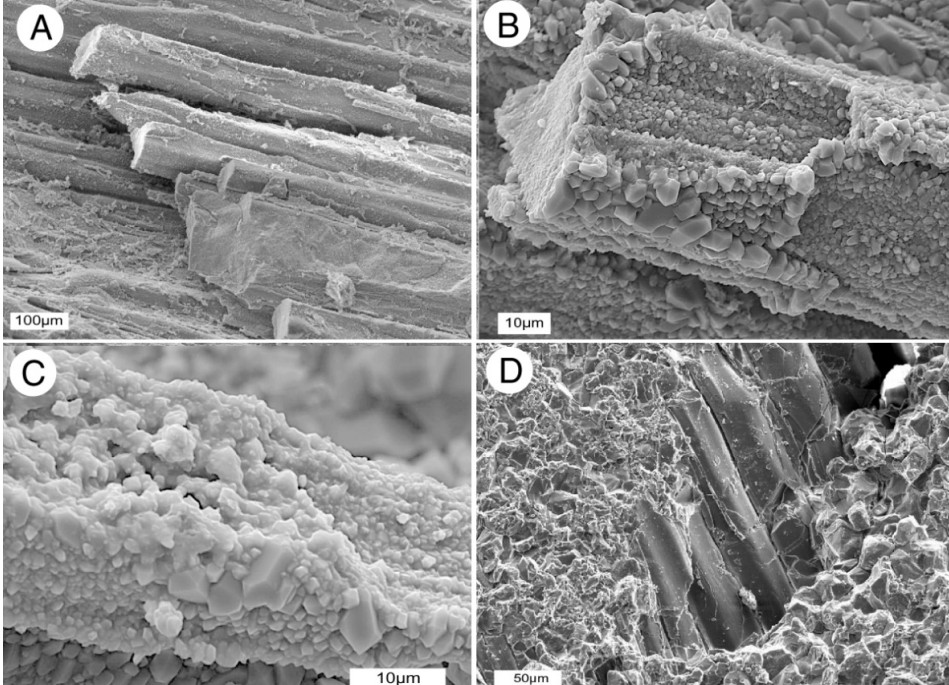

**Figure 12.** SEM views of periderm cells showing silica mineralization. (**A,B**) Oblique longitudinal views show cells are solidly replaced by quartz, with some intracellular spaces remaining empty. (**C**) High magnification shows that cells are encrusted with euhedral quartz crystals. (**D**) Smooth-surfaced cells are enclosed by a matrix of granular quartz crystals.

## 6.5. Macrophotography

The largest quartz crystals are located in open spaces created by wood decay (Figure 13).

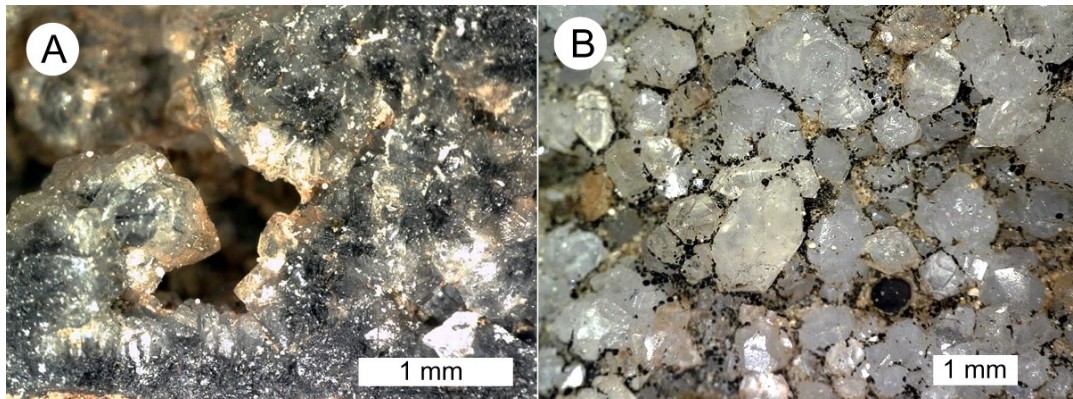

**Figure 13.** Euhedral quartz crystals in cavities created as decayed spaces in periderm tissue. (**A**) Decay space contains a geode-like lining, with the central space remaining empty. (**B**) Quartz crystals, formed on spaces in a rotted area, include double-terminated habits.

## 6.6. X-Ray Diffraction Evidence

Microcrystalline quartz and chalcedony are the only minerals observed by optical microscopy of thin sections. X-ray diffraction patterns (XRD) (Figure 14) likewise show quartz/chalcedony as the only detectable constituents (the two silica polymorphs are indistinguishable by XRD).

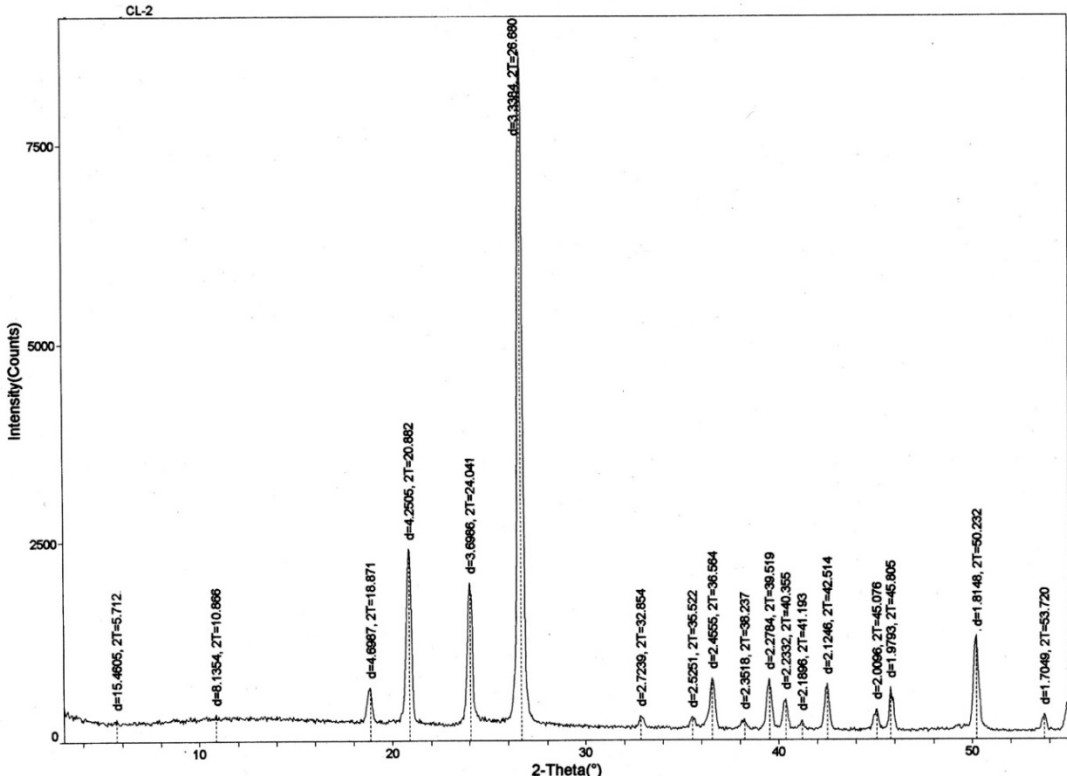

**Figure 14.** X-ray diffraction pattern for silicified lycopsid wood. All diffraction peaks are produced by quartz/chalcedony.

### 6.7. SEM/EDS Data

X-ray fluorescence spectra show $SiO_2$ as the primary constituent of Colorado lycopsid wood. The amount of relict carbon is variable, being highest in dark-colored specimens (Figure 15).

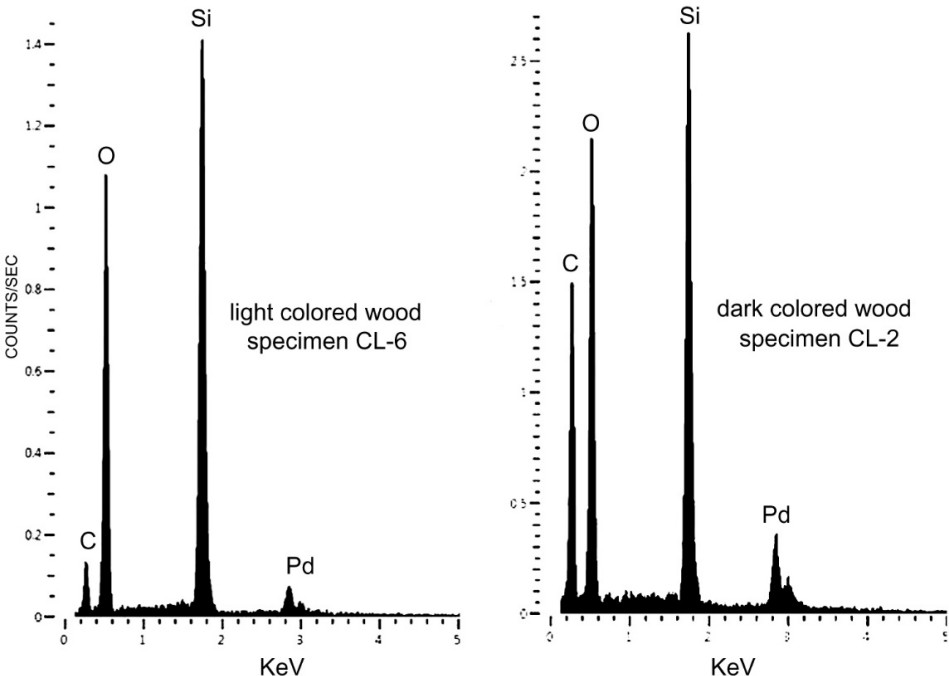

**Figure 15.** SEM/EDS spectra of lycopsid wood. (**A**) Light-colored specimen contains a small amount of relict carbon. (**B**) Dark-colored wood contains high levels of relict carbon. Pd peaks are from metal sputter-coated on the specimens to provide electrical conductivity.

## 7. Discussion

### 7.1. Mineralogy

In 1982, C.L. Stein used thirty-three samples of silicified wood representing different locations and ages to study the occurrence of silica in petrified wood. The published study [23] included three photos of a specimen of *Lepidodendron johnsonii*(= *Lepidophloios johnsonii* (Arnold) DiMichele) from the Trout Creek locality. The caption for a 110X image describes "poor preservation of scalariform tracheids by quartz". Two higher magnification views show euhedral quartz crystals, similar to the images we show in Figure 12. Stein [23] made the generalization that quartz in silicified wood was typically a result of diagenetic transformation from an opaline precursor. This opal A → opal CT → quartz transformation sequence is well-documented for the diagenesis of siliceous marine sediments and for siliceous hot spring sinter, but evidence from fossil wood is less clear. This sequence is certainly a possibility for some fossil wood occurrences, but many investigations [24–28] have shown that the fossilization of wood may involve multiple episodes of mineralization, and that quartz may occur as a primary precipitate, akin to the formation of crystalline quartz in veins and geodes.

The petrography of Colorado lycopsid wood is indicative of multiple episodes of silica mineralization. The first step appears to have been the precipitation of amorphous silica at cell walls via organic templating, when lignin and cellulose provided active sites for the attachment of monosilicic acid molecules from groundwater [29]. Most of the original organic matter was eventually lost, and the amorphous silica (opal) was eventually transformed to quartz during diagenesis. This transformation is supported by the absence of opal wood in Mesozoic deposits, compared to the abundance of opal wood in Cenozoic deposits. A second step in the petrifaction process was the deposition of chalcedony/quartz in the cell lumina. The coarse crystallinity of this filling material (Figure 10B,C) suggests that this

silica was a direct precipitate, not a transformation of an amorphous precursor phase. Relatively large quartz crystals occur in rot pockets, fracture zones, and some intercellular spaces, suggesting that the final phases of mineralization occurred during a later time, when lowered concentrations of dissolved silica allowed well-ordered lattices to develop in larger spaces within the wood (Figure 11).

Because of the Pennsylvanian age and the complex tectonic history of the region, the source of the dissolved silica is a matter of speculation. As a general rule, silicification of wood occurs from alkaline conditions, where silica is dissolved from siliceous biogenic sediment, glass-rich volcanics, or feldspar-rich host materials. Airborne dust has been proposed as a source of silica for chert associated with Paleozoic carbonates [30,31]. This eolian deposition is assumed to have been related to the increasing aridity. The dissolution of quartz was favored by the very small grain size, which produced a relatively large surface area, and the abrasion of grain surfaces that produced amorphous silica [31]. The coastal lowland swamp forests that are represented by the Trout Creek fossil site may not have been associated with the delivery of eolian dust, but extensive silicification of the Belden Shale host sediment makes evident that dissolved silica was readily available. The underlying Lower Mississippian Leadville Limestone is also locally silicified. As noted earlier, fossil specimens are predominantly angular wood fragments rather than intact stumps or logs. The external form of these specimens commonly preserve bark impressions (Figure 3) but the internal structure is often chaotic; degraded wood fragments lie in a horizontal layer at the bottom of a chamber that is filled with fine-grained quartz (Figure 8C,D).

### 7.2. Paleoenvironment

During the early Pennsylvanian, the Ancestral Rocky Mountains were at a much lower latitude than today (Figure 4), indicative of a warm, relatively seasonless climate. The low elevation environment favored the existence of extensive swamps, where lycopsids were the dominant trees. The coastal setting is evidenced by the transgressive/regressive nature of the Carboniferous sediments; the Belden Shale includes strata that contain marine invertebrates, in addition to continental sediments that preserve leaf and wood fossils.

Botanical aspects of lycopsid forests have received varied interpretations. Arborescent lycopsids may have had growth histories different from modern trees. Lycopsid trunks were characterized by secondary xylem that provided efficient water transport, but the thick secondary cortex (periderm) provided the structural support necessary to support large tree height. Some investigators have assumed that arborescent lycopsids were fast-growing [32–34], with growth rates perhaps 20 times faster than modern angiosperms and with maximum lifespans of 10–15 years [33]. An alternate interpretation is that the growth rates of arborescent lycopsids were very slow, with lifetimes of centuries [35]. The Trout Creek lycopsid fossil locality does not provide evidence to resolve this controversy, but the site does provide some clues with regards to the paleoenvironment. In other Pennsylvanian coal swamps, *Lepidodendron* plants were commonly destroyed by floods or storms, followed by recolonization by a new generation of trees [36]. This model may account for the fossils found at the Colorado site. Arnold's original reconstruction [2] depicted many standing trees, as well as intact fallen trunks, in swampy lowland fed by sluggish streams that carried fine sediment (Figure 16). However, we observed that the most abundant specimens are angular wood fragments, preserved as casts that contain degraded fragments that settled as horizontal layers prior to silicification. This  taphonomy suggests that the ancient swamp contained fallen trees, commonly experienced rapid decay, and that fragments were perhaps fluvially transported. These characteristics suggest an environment that was more chaotic than the calm scene portrayed in Arnold's sketch. The most chaotic possibility is the scenario proposed in 2015 by Thomas and Cleal [36], where Carboniferous swamp forest deposits were interpreted to have resulted from catastrophic storm events where cyclone-strength winds caused extensive damage to vegetation, accompanied by flooding with sediment-laden water. In this situation, the fossiliferous strata may represent a very brief depositional episode, regardless of the age of the forest trees.

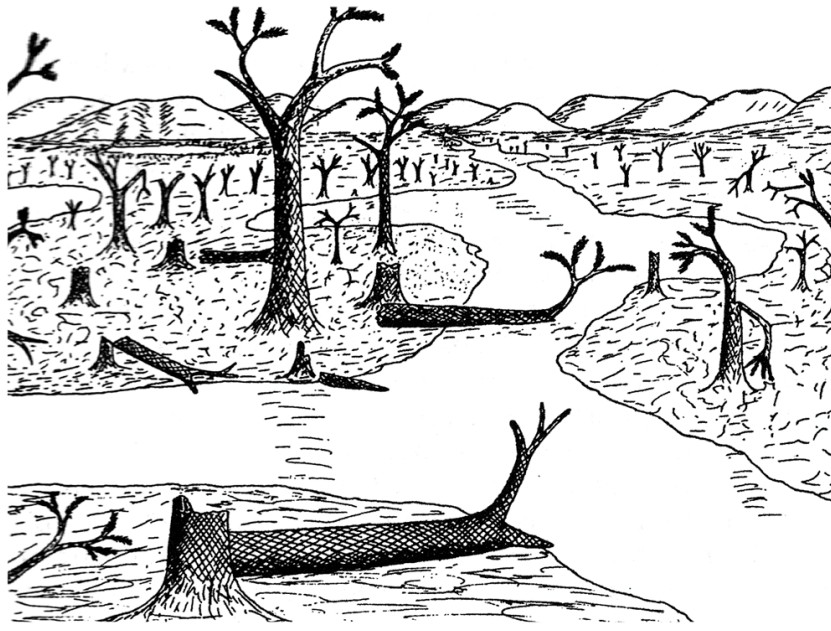

**Figure 16.** Arnold's 1940 reconstruction of the Trout Creek lycopsid forest [2].

## 8. Conclusions

Silicified plant material making up the Trout Creek lycopsid fossil locality in the Mosquito Range of south-central Colorado represents an unusual mode of preservation for a Pennsylvanian aged deposit. A single scientific study of the site in 1940 included the naming of a new species of lycopsid tree, *Lepidodendron johnsonii* (= *Lepidophloios johnsonii* (Arnold) DiMichele). Newly collected specimens provided an opportunity to investigate the mineralization sequence of silicified plant fossils from this unusual site.

The Trout Creek lycopsid fossil locality is inferred to represent a costal lowland swamp growing at low latitudes during the Pennsylvanian subperiod. Subsequent plate movements, mountain building, weathering and erosion placed the Trout Creek lycopsid fossil site in its present-day mountain meadow. Remnants of the ancient forest are limited to fragmental fossil plant remains, of which lycopsids are the only recognized component. The pith and most of the cortex of the buried stems became decayed. Only sections of small central axes and some periderm remained to become silicified. The mineralization of fossil wood at the Trout Creek lycopsid locality occurred in multiple stages. The anatomically preserved cell structure provides evidence of a templating process, where amorphous silica precipitated on cell walls. This silica was later transformed to fine-grained quartz during diagenesis. Layers of larger euhedral quartz crystals, growing over small quartz crystals along cell wall surfaces, represent further episodes of mineralization, when dissolved silica concentrations were lower. The largest euhedral quartz crystals are found in decayed spaces, representing the final episode of silicification. Loss on Ignition values indicate that most of the original organic matter was lost during fossilization, but X-ray fluorescence spectra show a relationship between relict carbon content and specimen color, with higher carbon values in dark-colored specimens.

**Supplementary Materials:** The following are available online at http://www.mdpi.com/2076-3263/9/12/510/s1.

**Author Contributions:** Conceptualization, M.V.; Investigation, M.V., G.E.M. and R.D.H.; Methodology, G.E.M.; Writing—original draft, G.E.M.; Writing—review & editing, M.V., G.E.M. and R.D.H.

**Funding:** This research received no external funding.

**Acknowledgments:** Specimens for this study were collected under the provisions of Colorado State Land Board Department of Natural Resources Temporary Access Permit Application No. 112723. We thank William DiMichele for his help for improving the final draft, and for the constructive suggestions of two anonymous reviewers.

**Conflicts of Interest:** The authors declare no conflicts of interest.

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
