# Peer review of "A Silicified Carboniferous Lycopsid Forest in the Colorado Rocky Mountains, USA"

_geosciences, doi:10.3390/geosciences9120510_

Round 1
Reviewer 1 Report
Please see attachment

Author Response
Thanks for your insights and constructive suggestions. We are happy to accommodate all of them. We have also added the several references that you cited. The various word spacing errors were inadvertently created when the editors converted the submitted text from left justified to full justification. We've repaired those glitches, and will watch for them in case they crop up again in the galley proof. However, the Geosciences copy editors are excellent and they strive for perfection.
Reviewer 2 Report
This is a good study and worth publishing, but will benefit from minor changes in focus. The long discussion of tectonic history is distracting and should be largely cut to improve the study. Correspondingly, the matter of dark coloration of cell walls will benefit from further discussion and the matter of source and mechanism of silica emplacement deserves more discussion.
Regarding dark cell walls, there is the statement that dark coloration of cell walls in the silicified wood is expected to be due to presence of carbon, but the possibility that it results from emplacement of small amounts of amorphous Fe/Mn-bearing compounds during diagenesis is not considered. It should be. The absence of considering the source of dissolved silica and the processes of silica transport to the lycopod specimens is quite noticeable. The mechanics of silica replacement can be induced by alkaline conditions acquiring dissolved opaline biosilica or altering volcanic ash during flushing with fresh waters within siliciclastic sediment. It is a common mechanism for silicification in younger sediments. Some comment on this possible mechanism would be appropriate for this study.
Specific comments:
line 11: Lepidodendron johnsonii is not a single word name. Correct this. Also line 38, 46, 176, 319
line 11 and many others: problem of spacing between words - a problem throughout the manuscript
line 33: use correct spacing before and after taxon names
line 42: Johnson discovered an occurrence of fossil Lepidodendron, not necessarily a forest
line 59-61: The extended discussion of tectonic uplift of the Rocky Mountains is not important to this study
line 77 and continuing: the discussion of geologic and tectonic history of the Ancestral Rocky Mountains and the Rocky Mountains is very long and there is no immediate indication of how it relates to understanding the silicification of lycopod wood that is the focus of this study. A focus on depositional environments of the Belden Formation would be relevant, but the tectonics discussion is distracting.
line 138 - 150: the stratigraphic information has much more relevance to the study than the tectonics
line 174: this is a contradictory sentence. Acetate peels are prepared with the use of hydrochloric acid to dissolve carbonate; use of hydrofluoric acid is standard in dissolution of silica, so it would obviously destroy silica-replaced cell walls. Rewrite
line 194: the conditions of silica replacement make me wonder if Arnold was correct in identifying Stigmaria rootlets as the source of the rounded silica masses in the xylem tissue
line 216, Fig 8C,D: the interpretation that matrix between collapsed wood fragments is entirely the product of precipitated microcrystalline quartz needs better documentation; the illustration provide little support for that determination
line 218, Fig 8E,F: same comment as above
line 355: the environment discussion is weak and would be more effective if shortened
line 383 to 391: much of this could be trimmed or cut
Terminology: The repeated use of the term “Lycopod forest” implies more certainty than is demonstrated by documentation. Lycopod trees may have a dominant taxon of the flora, but there is no evidence showing that they were the only common taxon. As mentioned in lines 54 to 58, occurrence could be the result of drift accumulation. While it is reasonable to infer they grew in a swamp environment, that is not supported with enough evidence to be something other than inference. To use these terms properly, the authors need to state that the occurrence is assumed to be a lycopod-dominated flora in a swamp environment, not declare it as a certainty. Use of the term “forest” is exaggeration and a thoughtless repetition of other authors use of the term. However, I would not deny them the right to use the term even if I think it inappropriate.
Author Response
Thanks for the careful review, and the useful suggestions for improving the manuscript. We have paid close attention to your advice, and adopted most of the proposed alterations. Here are point by point discussions:
Word spacing errors. These were unfortunately created when the editors changed the left justified text of the submitted manuscript to full justification prior to sending the manuscript out for review. We have repaired those glitches, Geoscience copy editors are always dedicated in their efforts to produce perfect galley proofs, but the review editing is a bit more relaxed. Dark cell walls. We have added the observation that SEM/XRF spectra do not show detectable elements such as Fe and Mn. Only Si, O, and C are observed to be present in the silicified walls.We've also clarified the mention of silicification processes. "Fossil Forest" usage: We have largely eliminated this terminology, substituting "fossil site", "fossil locality", etc. The "lycopod fossil forest" wording in the title is consistent with the observation that all of the fossils known from this locality are Lepidodendron, it is not merely a dominant taxon, but seemingly the only one. Tectonic history section: We have reduced the length of this section a bit, but we feel that this section is important. Merely focusing on the stratigraphy does not account for the fact that the fossil site has been moved a long distance in latitude and in elevation. Also, the stratigraphic section contains many units that are separated by faults or unconformities, and the observed sequence is closely related to tectonic history. HF and acetate peels: thanks for catching this cloudy description. It has been reworded for clarity. Stigmaria: We agree, Arnold's description of Stigmaria rootlets does not seem convincing, and we did not observe similar features in our recently collected specimens. This text has been reworded. Environment section: We have done some rewriting, and also eliminated one figure.